# Distinct Gut Microbiota Signatures Are Associated with Severity of Metabolic Dysfunction-Associated Steatotic Liver Disease in People with HIV

**DOI:** 10.3390/ijms26178165

**Published:** 2025-08-22

**Authors:** Riccardo Righetti, Felice Cinque, Bertrand Lebouché, Luz Ramos Ballesteros, Jean-Pierre Routy, Marina B. Klein, Jason Szabo, Joseph Cox, Julian Falutz, Louis-Patrick Haraoui, Cecilia T. Costiniuk, Alexandra De Pokomandy, Thomas Pembroke, Marco Constante, Manuela Santos, Giada Sebastiani

**Affiliations:** 1Chronic Viral Illness Service, Department of Medicine, McGill University Health Centre, Montreal, QC H4A 3J1, Canada; riccardorighetti1994@gmail.com (R.R.); jean-pierre.routy@mcgill.ca (J.-P.R.); cecilia.costiniuk@mcgill.ca (C.T.C.);; 2Infection and Immunity in Global Health Program, Research Institute of the McGill University Health Centre, Montreal, QC H4A 3J1, Canada; 3Internal Medicine Unit, Department of Medical and Surgical Science for Children and Adults, Azienda Ospedaliero-Universitaria Policlinico di Modena, University of Modena and Reggio Emilia, 41125 Modena, Italy; 4 Department of Microbiology and Infectious Diseases, Université de Sherbrooke, Sherbrooke, QC J1K 2R1, Canada; 5School of Medicine, Cardiff University, Cardiff CF14 4EP, UK; 6Department of Medicine, McMaster University, Hamilton, ON L8S 4L8, Canada; 7Department of Medicine, Université de Montréal, Montreal, QC H3C 3J7, Canada

**Keywords:** microbiome, liver fibrosis, metabolic dysfunction-associated steatohepatitis, liver stiffness measurement, cytokeratin-18

## Abstract

The progression of metabolic dysfunction-associated steatotic liver disease (MASLD) to severe forms, including metabolic dysfunction-associated steatohepatitis (MASH) and liver fibrosis, involves metabolic dysfunction, genetics, and gut dysbiosis. People with HIV (PWH) represent a high-risk group for MASLD, but the role of gut microbiota alterations in disease severity within this population remains poorly understood. We prospectively recruited PWH with MASLD, defined as the controlled attenuation parameter (CAP) ≥ 238 dB/m, and excluded those with viral hepatitis coinfection or alcohol abuse. Severe MASLD was defined as the presence of MASH (cytokeratin-18 ≥ 130.5 U/L) and/or significant liver fibrosis (liver stiffness ≥ 7.1 kPa). Stool samples were collected for 16S rRNA gene sequencing to characterize gut microbiota composition. Functional predictions were generated using PICRUSt. The differential abundance of bacterial taxa and predicted functions were analyzed using a generalized linear model with a negative binomial distribution. Among 34 PWH with MASLD, 18 (53%) met the criteria for severe MASLD. Microbiota profiling revealed significant differences in bacterial genera between the PWH with and without severe MASLD. Enrichment was observed in the *Ruminococcus gnavus* group, *Negativibacillus*, *Holdemanella*, *Subdoligranulum*, the *Eubacterium hallii* group, and *Butyricicoccus*, while depletion was seen in *Prevotella*, *Alloprevotella*, *Dialister*, *Catenibacterium*, the *Christensenellaceae R 7* group, *Clostridium sensu stricto*, *Olsenella*, *Oscillospiraceae UCG-005*, *Libanicoccus*, and the *Eubacterium siraeum* group. Predicted functional pathways related to fatty acid degradation, folate biosynthesis, and amino acids metabolism did not differ between groups. MASLD severity in PWH is associated with a distinct gut microbiota signature, though not with functional pathway alterations. Microbial profiling may complement existing non-invasive biomarkers for risk stratification in this high-risk population.

## 1. Introduction

The advent of antiretroviral therapy (ART) for people with HIV (PWH) has transformed their condition into a chronic, manageable condition [1]. Chronic liver disease remains a significant cause of non-AIDS-related mortality in PWH [2], with the primary etiology shifting from viral hepatitis to metabolic dysfunction-associated steatotic liver disease (MASLD) [3]. MASLD ranges from a relatively benign accumulation of triglycerides in the hepatocytes, to metabolic dysfunction-associated steatohepatitis (MASH), a severe liver condition characterized by inflammation leading to liver fibrosis and cirrhosis [4,5]. The increased severity of this condition in PWH is driven by the higher occurrence of traditional metabolic risk factors, alongside HIV-specific contributors such as chronic inflammation, ART-induced mitochondrial damage, and lipodystrophy [6]. Thus, clarifying the underlying processes that drive MASH and liver fibrosis in PWH is pivotal to improving both diagnostic and therapeutic strategies.

Liver cross-talk with other organs, particularly the gut–liver axis, has recently been recognized as a critical factor in MASLD progression [7]. Notably, the gut microbiota is the primary component driving the dynamics of the gut–liver axis. Reduced gut microbiota diversity in PWH may elevate the risk of liver injury driven by disruptions in the gut–liver axis, ultimately contributing to significant liver fibrosis [8]. Recent findings indicate that *Ruminococcus*, *Streptococcus*, *Holdemanella*, *Blautia*, and *Lactobacillus* are more abundant in PWH who have MASLD compared with those without the liver condition [9]. Another study in PWH linked steatosis to depletions in *Akkermansia muciniphilia* and *Bacteroides dorei*, along with an increased abundance of *Prevotella copri*, *Finegoldia magna*, and *Ruminococcus bromii* [10]. Functional analysis has also identified abnormalities in microbiome-expressed metabolic pathways among MASLD patients without HIV [11]. Excessive or decreased short-chain fatty acid production [12] as well as amino acids [12,13] and folate depletion [13,14] are related to MASH progression due to the depletion of glutathione, the liver’s key intracellular antioxidant. However, the role of gut bacterial imbalances in metabolic pathways in MASLD in PWH remains unclear.

In this study, we aimed to characterize the microbiota signatures and investigate the alterations in enzymatic pathways associated with severe MASLD, defined as the presence of MASH or significant liver fibrosis, compared with non-severe MASLD, in PWH with no viral hepatitis coinfection or significant alcohol intake.

## 2. Results

Among 110 individuals screened from the LIVEHIV cohort, 34 were enrolled after applying the exclusion criteria (Figure 1). The characteristics of the study population stratified by severe MASLD status are presented in Table 1. Overall, the mean age was 51 years (standard deviation of 10), the majority were male, and the most represented ethnicities were White and Hispanic. Sixteen (47%) patients were overweight and twelve (35%) were obese. MASH diagnosed by CK-18 was present in 14 (41%) patients, while 8 (24%) had significant liver fibrosis by LSM. A total of 18 (53%) PWH were classified as severe MASLD. Compared with those with non-severe MASLD, PWH with severe MASLD had a higher absolute CD4 cell count, higher ALT, and higher triglycerides and fasting glucose. They also presented with higher CAP values. As expected, patients with severe MASLD had higher LSM and CK-18 values compared with those with non-severe MASLD.

### 2.1. Microbiota Diversity and Taxonomic Differences

We performed alpha- (Chao1, Shannon’s, and Phaith’s Diversities) and beta-diversity (Aitchison’s distance) analyses but the analyses were not significant. Subsequently, we analyzed genus-level differences between the two study groups and identified several significant variations (Figure 2). Within the phylum Firmicutes, patients with severe MASLD showed a lower abundance of *Senegalimassilia*, the *Eubacterium ruminantium* group, *Dialister*, *Catenibacterium*, *Kandleria*, the *Family XIII AD3011* group, *Asaccharospora*, the *Christensenellaceae R 7* group, *Clostridium sensu stricto*, *Oscillospiraceae UCG-005*, the *Ruminococcaceae NK4A214* group, *Libanicoccus*, *Romboutsia*, the *Eubacterium siraeum* group, *Megasphaera* and a higher abundance of the *Ruminococcus gnavus* group, *Negativibacillus*, the *Eubacterium eligens* group, *Anaerostipes*, *Holdemanella*, *Subdoligranulum*, the *Eubacterium hallii* group, and *Butyricicoccus*. Among Bacteroidetes, a lower abundance of *Prevotella 7*, *Phascolarctobacterium*, *Alloprevotella*, *Prevotella 9*, and *Prevotella* were observed in severe MASLD, whereas *Bacteroides* was more abundant. Within less represented phyla, *Desulfovibrio*, *Slackia*, *Olsenella*, and *Lutaonella* were also reduced in severe MASLD.

### 2.2. Microbial Functional Pathways

Functional profiling revealed no significant MASLD group-level differences in the overall KEGG pathways for fatty acid degradation, amino acids metabolism, or folate biosynthesis pathways. However, the enzyme-level analysis identified the differential expression of several microbial genes. In the severe MASLD group, we observed a statistically significant higher predicted presence of succinate semialdehyde dehydrogenase, L-asparaginase beta-aspartyl peptidase, alanine dehydrogenase, aspartate carbamoyltransferase regulatory subunit, and alanine dehydrogenase (K19244), alongside a lower predicted presence of alanine dehydrogenase (K00259), glutamate dehydrogenase (NAD(P)+), and 1-pyrroline-5-carboxylate dehydrogenase (Figure 3a). Regarding fatty acid metabolism (Figure 3b), the predicted presence of 3-hydroxyacyl-CoA dehydrogenase, rubredoxin NAD reductase, and aldehyde dehydrogenase (NAD+) was significantly higher in severe MASLD. In contrast, the predicted presence of glutaryl-CoA dehydrogenase was significantly lower. Regarding folate biosynthesis (Figure 3c), severe MASLD was associated with a higher predicted presence of D-erythro-7,8-dihydroneopterin triphosphate epimerase, dihydromonapterin reductase, dihydroneopterin triphosphate diphosphatase, dihydroneopterin aldolase, and 4a-hydroxytetrahydrobiopterin dehydratase. Conversely, lower predicted levels were observed for para-aminobenzoate synthetase (K03342 and K13950).

### 2.3. Integration with Clinical Features

To assess whether specific microbial signatures or enzymatic pathways correlated with clinical features, we conducted exploratory correlation analyses. The expression of dihydroneopterin triphosphate diphosphatase (K08310) moderately correlated with LSM (Appendix A). The expression levels of microbial enzymes—including 7-cyano-7-deazaguanine reductase (K06879), succinate semialdehyde dehydrogenase (K08324), 3-hydroxyacyl-CoA dehydrogenase/enoyl-CoA hydratase/3-hydroxybutyryl-CoA epimerase/enoyl-CoA isomerase (K01825), GTP-cyclohydrolase I (K01497), erythron-7,8-dihydroneopterin-triphosphate epimerase (K07589), and dihydromonapterin reductase/dihydrofolate reductase (K13938)—correlated with serum triglyceride levels (Appendix A). Similarly, the abundance of the bacterial genera *Holdemanella* and *Escherichia-Shigella* was associated with serum triglycerides (Appendix A). These findings suggest possible links between the microbiome and hepatic or metabolic parameters. We further constructed multivariable models incorporating microbial genera, enzyme profiles, and clinical covariates. However, no statistically significant associations were retained after correction for multiple comparisons, likely due to sample size limitations and inter-individual variability.

## 3. Discussion

This study highlights the potential role of the gut microbiome in contributing to significant liver fibrosis and necroinflammation in MASLD, providing valuable insights into the underlying pathophysiology in PWH. Our study builds upon and extends the work of Martínez-Sanz et al. [9] and Yanavich et al. [10] by focusing on the association between the microbiota signature and MASLD severity. Importantly, we employed innovative techniques, with the use of CK-18 as a non-invasive biomarker of hepatocellular apoptosis, and emphasized alterations in the metabolic pathways known to play a pathogenic role, offering new insights into disease mechanism. Previous studies have reported that CK-18 achieves a sensitivity of 92.9% and specificity of 63% for MASH diagnosis [15], while FibroScan^®^ has demonstrated a reliable accuracy in diagnosing significant fibrosis in PWH when compared with paired liver biopsies [16].

Within the phylum Firmicutes, the severe MASLD group exhibited decreased levels of bacteria recognized as protective factors against MASLD, including *Senegalimassilia* [17], *Dialister* [9], *Catenibacterium* [9], the *Christensenellaceae R 7* group [18], *Clostridium sensu stricto* [19], and *Oscillospiraceae UCG-005* [20]. Moreover, we found an enrichment of the genera the *Ruminococcus gnavus group* [21], *Negativibacillus* [22], *Holdemanella* [9], *Subdoligranulum* [23], *Eubacterium hallii* group [24], and *Butyricicoccus* [23,25] which are associated with MASLD and visceral fat accumulation. Among taxa correlated with serum triglycerides, we observed *Holdemanella* and *Escherichia–Shigella*, consistent with prior associations to inflammation and lipid metabolism. Contrary to prior studies, we observed a higher abundance of the *Eubacterium eligens* group, which has been associated with a reduction in visceral and subcutaneous fat [26], and *Anaerostipes* which is believed to be protective against MASLD due to butyrate production [27], but has also been associated with diabetes [28]. This unexpected enrichment may reflect functional divergence at the strain level or altered metabolic output in the context of HIV infection. Furthermore, we observed a reduced abundance of *Megasphaera*, previously linked to MASLD [29,30,31,32], along with *Romboutsia*, although its association with MASLD and disease severity remains inconclusive. These discrepancies may stem from differences in our study population, which is Canadian and likely adheres to a Western diet in a manner distinct from the populations in the cited studies. Additionally, unlike the referenced articles, our study included patients with HIV, a condition known to influence human microbiota composition [33]. PWH exhibit impaired α-diversity and unique microbiota signatures at the genus level [34]. Notably, this condition is characterized by an enrichment of Gammaproteobacteria and a concomitant depletion of beneficial taxa [35]. Consistent with previous findings, the severe MASLD group showed a reduced abundance of Bacteroidetes genera associated with HIV, such as *Prevotella* and *Alloprevotella*, which appear to have a protective role against MASLD [9]. *Phascolarctobacterium* levels were reduced in severe MASLD, in contrast to a previous study that associated this genus with increased liver stiffness [36]. Among the Bacteroidetes genera, *Bacteroides* was the only one found to be more prevalent, although its role in MASLD remains controversial [37]. Regarding less represented phyla, we found a decreased abundance of *Desulfovibrio*, *Slackia*, *Olsenella*, and *Lutaonella*. While *Olsenella* has been reported to exert a protective role in MASLD [38], findings related to *Desulfovibrio* remain discordant. The literature on the genera *Slackia* and *Lutaonella* remains scarce, but the former has been associated with lean MASLD in a single study [39].

Regarding amino acid metabolism, a decreased biosynthesis of alanine, aspartate, and glutamate has been described in PWH [40]. We found no significant differences between MASLD severity groups, but interestingly observed a predicted higher expression of L-asparaginase beta-aspartyl peptidase (K01251) in the severe MASLD group. This enzyme catalyzes the conversion of the amino acid L-asparagine into aspartate and ammonia [41]. It is utilized in chemotherapy for acute lymphocytic leukemia and has been linked to hepatic steatosis in a dose-dependent manner, due to the depletion of asparagine and arginine which may result from associated arginase activity [42]. This leads to persistent hepatic steatosis with minimal inflammation and necrosis, resembling the steatosis seen in Kwashiorkor and derived from arginine depletion [42]. These findings are concordant with previous literature, showing significantly reduced arginine levels in plasma in severe fibrosis compared with mild/moderate stages [43]. Concordantly, the expression of 1-pyrroline-5-carboxylate dehydrogenase (K00266) was estimated as being diminished in our patients with more severe disease, supporting the hypothesis of impaired antioxidant capacity. This enzyme is involved in the redox balance and the proline degradation pathway, converting proline into glutamate [44].

Concerning fatty acid metabolism, we could not find evidence for reduced degradation levels in severe MASLD. Conversely, several enzymes were predicted as upregulated, including *3-hydroxyacyl-CoA dehydrogenase (K00074)*, *rubredoxin NAD_+_ reductase (K05912)*, and *aldehyde dehydrogenase NAD_+_ (K00128)*, suggesting increased microbial fatty acid degradation. This may represent a maladaptive feature of this population, more exposed to dietary fatty acids and host lipid burden.

The analysis of the microbiome folate pathway did not show the anticipated reduction in biosynthesis expression in the severe MASLD group. A decreased microbiota production of group B vitamins in PWH has already been described [40]. However, since determining the extent to which gut-derived vitamins influence the host’s systemic levels remains challenging and is diet-dependent, these results should be interpreted with caution. Interestingly, we observed a suggested increased expression of several enzymes involved in folate metabolism, including *GTP cyclohydrolase (K01497)*, *erythron-7,8-dihydroneopterin-triphosphate-epimerase (K07589)*, *dihydromonapterin reductase/dihydrofolate reductase (K13938)*, and *dihydroneopterin triphosphate diphosphatase (K08310)*, the latter of which was also correlated with liver stiffness. The role of folate in MASLD remains unclear, but has been linked to hyperhomocysteinemia [45], choline metabolism [46], AMPK pathway activation [47], and NADPH oxidase synthesis [48]. Several enzymes, including 7-cyano-7-deazaguanine reductase (*K06879*), dihydroneopterin triphosphate diphosphatase (*K08310),* and succinate semialdehyde dehydrogenase (*K08324*), were correlated with serum triglycerides, suggesting functional microbiome–host interactions in lipid regulation. While these correlations are exploratory and require validation, they may offer insights into early dysbiosis-associated mechanisms in MASLD progression.

We should also consider that certain ART classes and specific drugs have been associated with an increased risk of MASLD. For example, Bischoff et al. [49] reported progressive increases in hepatic steatosis among patients treated with integrase strand transfer inhibitors (INSTIs) or tenofovir alafenamide (TAF). However, other longitudinal studies did not observe a significant impact of these agents on liver fibrosis, suggesting that their contribution may be limited to steatosis rather than fibrogenesis [50]. Beyond hepatic effects, ART can also influence the gut microbiome, leading to both beneficial and adverse changes. While ART effectively suppresses HIV replication and restores immune function, it does not fully normalize gut health: persistent inflammation, microbial translocation, and the incomplete recovery of beneficial bacterial communities are frequently observed [51]. Importantly, different ART regimens appear to differentially affect the gut microbiome, and certain bacterial taxa associated with metabolic homeostasis may not recover under long-term ART. Taken together, these findings highlight that the interplay between ART, steatosis, dysmetabolism, and the gut microbiome remains incompletely understood and warrants further investigation. Given our limited sample size, we were unable to explore the association between ART and gut microbiome.

Our study has several limitations, including the lack of a control group without HIV and of a group of PWH without MASLD. Additionally, the relatively small sample size and the absence of histological confirmation, excluded due to its invasiveness, represent further limitations of the study. Another limitation is the lack of detailed information on patients’ dietary habits or supplement consumption, which may influence microbiota composition. Despite these constraints, this study provides valuable insights into the relationship between gut dysbiosis, metabolic pathways, and liver disease in PWH. To confirm the significance of microbiota signatures and metabolic alterations in this population, larger multi-center studies are warranted.

In conclusion, our study suggests a possible link between gut dysbiosis and MASLD severity in PWH, highlighting metabolic pathway alterations potentially linked to disease progression. These insights offer a new perspective on MASLD in PWH, paving the way for future research into the pathogenic mechanisms of dysbiosis.

## 4. Materials and Methods

### 4.1. Study Design and Participants

This is a single-center cross-sectional study conducted at the Chronic Viral Illness Service of the McGill University Health Centre (MUHC). From September 2017 to December 2024, all participants were enrolled in conjunction with the prospective clinical cohort LIVEr disease in HIV (LIVEHIV) [52]. Consecutive patients without viral hepatitis coinfection or antibiotic use within the past two months were offered enrollment into this sub-study. Participants were included if they met the following inclusion criteria: (i) 18 years or older; (ii) HIV infection documented by positive enzyme-linked immunosorbent assay (ELISA) with Western blot confirmation; (iii) on stable treatment with ART with controlled infection (HIV RNA < 20 copies/mL) for at least 6 months; (iv) having a diagnosis of hepatic steatosis, defined as a controlled attenuation parameter (CAP) ≥ 238 dB/m [29,53]. The exclusion criteria were (i) other causes of chronic liver disease, including coinfection with hepatitis B or C virus (by serologies), significant alcohol consumption (defined as >21 drinks/week in men and >14 drinks/week in women [5]), hemochromatosis, Wilson’s disease, autoimmune hepatitis, or previous liver transplantation; (ii) a history of inflammatory bowel disease or bariatric surgery; (iii) malnutrition (defined as body mass index, [BMI] < 18.5 Kg/m^2^); (iv) treatment with antibiotics within 2 months before inclusion; (v) pregnancy; (vi) decompensated cirrhosis; (vii) hepatocellular carcinoma.

### 4.2. Ethics

This study was approved in 2017 by the Research Ethics Board of the Research Institute of the MUHC (2017-2926) and conducted in accordance with the declaration of Helsinki. All participants provided written informed consent prior to enrolment. The manuscript was prepared according to the STROBE statement checklist of items.

### 4.3. Transient Elastography Examination with CAP

During the study visit, a liver stiffness measurement (LSM) and CAP were acquired using vibration-controlled transient elastography (Fibroscan^®^, Echosens, Paris, France) on patients fasting for at least 3 h by an experienced operator. The M probe was routinely utilized, while the XL probe was employed if the M probe was unsuccessful, or the patient had a BMI exceeding 30 Kg/m^2^. A transient elastography examination was considered reliable if it met a minimum of 10 validated measurements and an interquartile range (IQR) below 30% of the median value [54].

### 4.4. Clinical and Biological Parameters

We collected data within 3 months from the transient elastography examination, namely demographic information, time since HIV diagnosis (defined as the interval between the date of patients’ first positive HIV test and the date of the visit), ART class (non-nucleoside reverse transcriptase inhibitors [NNRTIs], protease inhibitors [PIs], INSTI, TAF, nucleoside reverse transcriptase inhibitors [NRTIs] including stavudine and didanosine), BMI, liver serum biomarkers, lipid profile, and hematological and immune–virological parameters. Undetectable viral load was defined as HIV viral load < 20 copies/mL. Overweight and obesity status were defined as a BMI 25–29 Kg/m^2^ and ≥30 Kg/m^2^, respectively. Questionnaires, including the AUDIT-C for alcohol use and a dedicated questionnaire for drug use, were administered. Mild alcohol consumption was defined as any intake <21 drinks/week in men and <14 drinks/week in women. MASLD was defined as hepatic steatosis by CAP plus at least one of the following criteria: (i) overweight or obesity; (ii) previous diagnosis or treatment for type 2 diabetes; (iii) blood pressure ≥ 130/85 mmHg OR treatment for hypertension; (iv) triglycerides > 1.69 mmol/L or lipid-lowering therapy; (v) HDL cholesterol < 1.03 mmol/L (males) OR < 1.30 mmol/L (females) OR lipid-lowering therapy. Severe MASLD was defined as the presence of MASH, indicated by cytokeratin-18 (CK-18) > 130.5 U/L [15,55,56], and/or significant liver fibrosis, defined as LSM > 7.1 kPa [16].

Plasma and stool sample collection, microbial DNA extraction, amplicon library construction, and sequencing were also conducted.

Immediately after collection at the hospital, plasma samples were centrifuged and stored at −80 °C until used. Stools were collected by patients into RNA-Later stabilization solution (ThermoFisher Scientific, Montreal, QC, Canada) and stored at −80 °C within 7 h of collection. Plasma was used to measure CK-18 using a commercially available human cytokeratin ELISA kit (MJS Biolynx inc, Brockville, ON, Canada), according to the manufacturer’s protocol. Stool sample DNA was isolated at the laboratory of the Chronic Viral Illness Service of MUHC using the PowerSoil DNA isolation kit (MO BIO Laboratories, Carlsbad, CA, USA). Amplicon library preparation and sequencing for the stool microbiota analysis was then performed at the Genome Québec Innovation Center. Amplicon libraries were constructed with bacterial/archael PCR primers 347F and 803R, which target the V3–V4 region of the 16S rRNA gene. A second PCR was performed to add sample barcodes and the adaptor sequences used by the Illumina sequencing systems. Samples were normalized using the Quant-iT™ PicoGreen^®^ dsDNA Assay Kit (ThermoFisher Scientific, Montreal, QC, Canada), then pooled and purified using Agencourt AMPure beads (Beckman Coulter, Mississauga, ON, Canada). The library was then sequenced on the MiSeq system (Miseq v2 reagent kit, 500 cycles PE—Illumina, San Diego, CA, USA), spiked with 20% PhiX for quality control (Illumina, San Diego, CA, USA). Demultiplexed forward and reverse 16S rRNA gene sequences were provided by Genome Québec and aligned using the Paired-End Read merger (PEAR). To obtain the Amplicon Sequences Variants (ASVs), sequences were then processed in R, version 4.4.2, using the package Divisive Amplicon Denoising Algorithm 2 (DADA2) [57] and the SILVA reference database, version 138.1 [58]. FastTree 2 was used to calculate a phylogenetic tree of sequences [59] and data were explored using the phyloseq package [60]. To infer microbial functional profiles, predicted metagenomes were generated using Phylogenetic Investigation of Communities by Reconstruction of Unobserved States (PICRUSt) [61]. This tool infers the abundance of gene families, including metabolic enzymes, based on the 16S rRNA gene data. Predicted gene abundances were then mapped to the Kyoto Encyclopedia of Genes and Genomes (KEGG) Orthology identifiers and functionally annotated using the KEGG database [62]. The expression of metabolic enzymes pathways was inferred from these KEGG Orthology profiles, allowing for comparative analysis across study groups.

### 4.5. Statistical Analysis

Taxonomic differences were evaluated using a generalized linear model with a negative binomial distribution [63], adjusting for sex as a confounding variable. Statistical differences between groups were identified using estimated marginal means using the emmeans package [64]. A correlation analysis was performed using Pearson’s test on normally distributed data, or Spearman’s correlations elsewhere. Qualitative variables were compared using Fisher’s exact test and quantitative variable using Mann–Whitney’s test. All hypotheses’ tests were conducted at a 5% level of significance (2-sided).

## Figures and Tables

**Figure 1 ijms-26-08165-f001:**
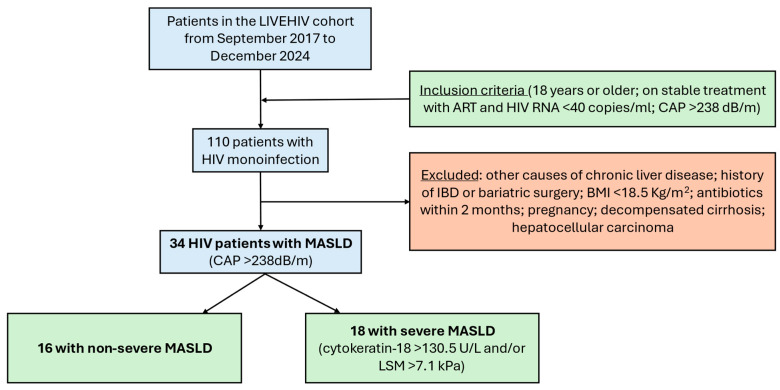
Flow chart displaying the selection of study participants. Legend: HIV, human immunodeficiency virus; CAP, controlled attenuation parameter; IBD, inflammatory bowel disease; BMI, body mass index; MASLD, metabolic dysfunction-associated steatotic liver disease; LSM, liver stiffness measurement.

**Figure 2 ijms-26-08165-f002:**
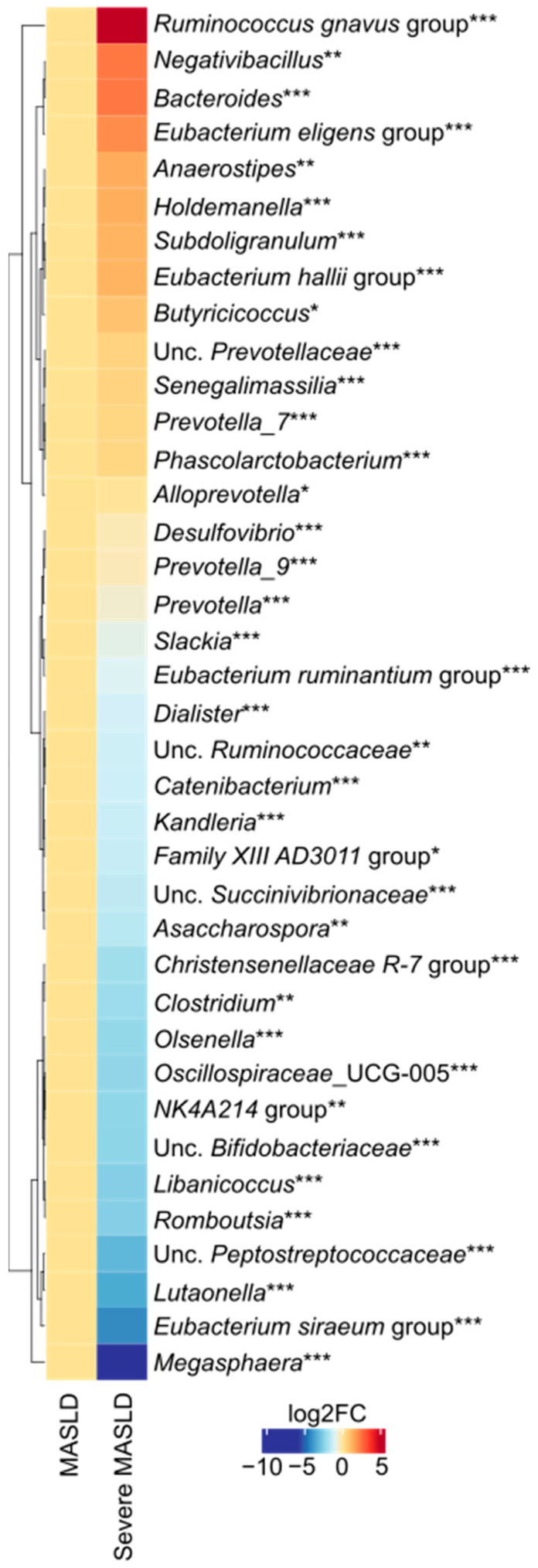
Heatmap illustration showing microbiome signature with significant differences between MASLD and severe MASLD groups, divided for study participants. Legend: MASLD, metabolic dysfunction-associated steatotic liver disease; sMASLD, severe metabolic dysfunction-associated steatotic liver disease; UCG, unclassified clostridia group; Unc, unclassified. * *p* < 0.05; ** *p* < 0.01; *** *p* < 0.001. *p*-values were not corrected for multiple testing.

**Figure 3 ijms-26-08165-f003:**
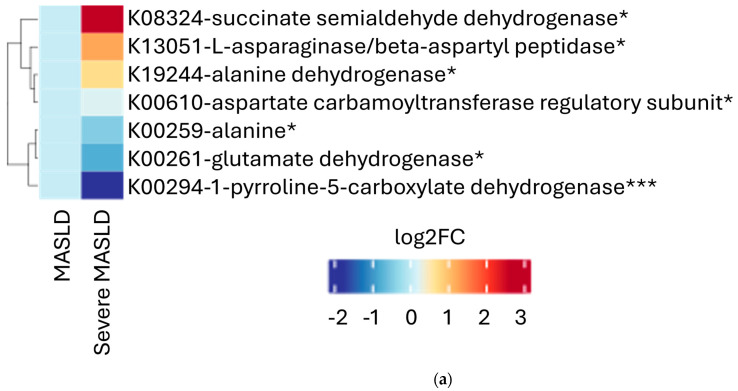
Heatmap illustrations of upregulated expression of specific enzymes in the severe MASLD and non-severe MASLD groups, divided by study participants. (**a**) Alanine, aspartate, and glutamate patterns. (**b**) Fatty acid metabolism. (**c**) Folate biosynthesis. Legend: MASLD, metabolic dysfunction-associated steatotic liver disease; sMASLD, severe metabolic dysfunction-associated steatotic liver disease; NADP, nicotinamide adenine dinucleotide phosphate; NAD, nicotinamide adenine dinucleotide; CoA, Coenzyme A; GTP, guanosine triphosphate. * *p* < 0.05; ** *p* < 0.01; *** *p* < 0.001.

**Table 1 ijms-26-08165-t001:** Characteristics of the study population by severe MASLD status (n = 34).

	Whole Study Population (n = 34)	Severe MASLD (n = 18)	Non-Severe MASLD (n = 16)	*p*-Value
**Age** (years)	51 (10)	53 (10)	49 (11)	0.293
**Male sex** (%)	31 (91)	17 (94)	14 (88)	0.476
**Ethnicity** (%)				
White	21 (62)	13 (72)	8 (50)	0.286
Hispanic	9 (26)	2 (11)	6 (38)	
Black non-Hispanic	3 (9)	1 (6)	2 (13)	
**MSM** (%)	21 (62)	12 (67)	9 (56)	0.533
**IDU** (%)	1 (3)	1 (6)	0	0.325
**Current alcohol use** (%)	20 (59)	11 (61)	9 (56)	0.773
**Current smoking** (%)	3 (9)	1 (6)	2 (13)	0.476
**Hypertension** (%)	11 (32)	7 (39)	4 (25)	0.387
**Diabetes** (%)	4 (12)	5 (28)	0	0.027
**History of cardiovascular event** (%)	4 (12)	4 (22)	1 (6)	0.189
**BMI** (Kg/m^2^)	28.9 (4)	29.9 (5)	28.0 (4)	0.222
**Time since HIV diagnosis** (years)	16.3 (7)	17.5 (8)	15.1 (7)	0.388
**CD4 cell count** (cells/μL)	581 (248)	680 (249)	488 (213)	0.028
**CD4/CD8 ratio**	1 (1)	1 (1)	1 (0)	0.714
**Current ART regimen** (%)				
NRTIs	28 (82)	13 (72)	15 (94)	0.100
NNRTIs	9 (26)	6 (33)	3 (19)	0.336
Protease inhibitors	11 (32)	8 (44)	3 (19)	0.097
Integrase inhibitors	22 (65)	10 (56)	12 (75)	0.236
**ALT** (IU/L)	27 (11)	32 (9)	23 (11)	0.014
**AST** (IU/L)	25 (9)	27 (8)	23 (10)	0.150
**GGT** (IU/L)	44 (24)	51 (26)	39 (22)	0.112
**Platelets** (10^9^/L)	215 (61)	220 (68)	210 (54)	0.676
**Bilirubin** (mmol/L)	13 (10)	15 (14)	10 (9)	0.171
**Albumin** (g/L)	43 (4)	44 (3)	42 (5)	0.888
**Creatinine** (mmol/L)	83 (14)	82 (13)	84 (15)	0.646
**Triglycerides** (mmol/L)	2 (2)	2 (2)	2 (1)	0.033
**Total cholesterol** (mmol/L)	5 (1)	5 (1)	5 (1)	0.370
**HDL** (mmol/L)	1 (0)	1 (0)	1 (0)	0.106
**LDL** (mmol/L)	3 (1)	2 (1)	3 (1)	0.099
**Fasting glucose** (mmol/L)	6 (1)	6 (1)	6 (0)	0.003
**Glycosylated Hemoglobin**	6 (1)	6 (1)	6 (0)	0.407
**Statin use** (%)	13 (38)	9 (50)	4 (25)	0.134
**LSM** (kPa)	7 (5)	9 (6)	5 (1)	0.007
**CAP** (dB/m)	300 (48)	319 (49)	283 (41)	0.031
**Cytokeratin-18** (U/L)	185 (161)	253 (177)	81 (26)	0.002

Legend: Continuous variables are expressed as mean (standard deviation) and categorical variables are expressed as frequencies (%). The p-values refer to Student’s *t*-test or χ^2^ test between severe MASLD and MASLD. Abbreviations: ALT, alanine aminotransferase; ART, antiretroviral therapy; AST, aspartate aminotransferase; BMI, body mass index; CAP, controlled attenuation parameter; GGT, gamma-glutamyl transpeptidase; HDL, high-density lipoprotein; HIV, human immunodeficiency virus; IDU, injection drug use; IUs, international units; LDL, low-density lipoprotein; LSM, liver stiffness measurement; MSM, men having sex with men; MASLD, metabolic dysfunction-associated steatotic liver disease; NNRTIs, non-nucleoside reverse transcriptase inhibitors; NRTIs, nucleoside reverse transcriptase inhibitors.

## Data Availability

The original contributions presented in this study are included in the article/Appendix A. Further inquiries can be directed to the corresponding author.

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
