# Peer review of "Distinct Gut Microbiota Signatures Are Associated with Severity of Metabolic Dysfunction-Associated Steatotic Liver Disease in People with HIV"

_ijms, 2025, doi:10.3390/ijms26178165_

Round 1
Reviewer 1 Report
Comments and Suggestions for Authors
This is an interesting and well-structured study investigating the association between gut microbiota composition, predicted functional pathways, and the severity of metabolic dysfunction-associated steatotic liver disease (MASLD) in a cohort of people with HIV (PWH). The research question is clinically relevant, as MASLD is an emerging cause of morbidity in this population.
Strengths:
- The title is clear, concise, and accurately reflects the study's content. The abstract provides a comprehensive summary of the study's background, methods, key results, and conclusion. The introduction effectively sets the stage by establishing the growing prevalence of MASLD as a significant comorbidity in PWH.
- The authors successfully identify a gap in the current literature—specifically, the need to understand the role of gut dysbiosis in MASLD
- The study design is clearly described, as are the results and the discussion
Weaknesses and Suggestions:
- The list of enriched and depleted bacterial genera in the abstract is quite long and detailed. While informative, it could be slightly streamlined for brevity. Consider highlighting only the most significant or well-known genera to improve readability.
- Sample Size: The most significant limitation of this study is the small sample of 34 participants (18 severe, 16 non-severe MASLD). This low number severely limits statistical power, increases the risk of Type II errors (failing to detect a true effect), and makes the significant findings susceptible to being false positives. Consider improving this aspect in the manuscript.
- Markers: The use of CK-18 and LSM as non-invasive surrogates for MASH and fibrosis is practical and avoids invasive biopsies. However, these are not the gold standard. A brief mention of the diagnostic accuracy or limitations of these markers in the methods or discussion would add valuable context.
- Functional Prediction: PICRUSt is a valuable tool for inferring function from 16S data, but it remains a prediction. This is an inherent limitation compared to direct functional measurement via metagenomics or metabolomics. The authors are careful to use terms like "predicted presence," which is appropriate. This point should always be emphasized when discussing the functional results.
- Dietary Data: The lack of information on participants' dietary habits is a major confounding factor in any microbiome study and is correctly identified as a limitation by the authors.
- Figure 2: The paper reports numerous bacterial genera as significantly different between groups. It is not explicitly stated whether the p-values for these individual taxonomic comparisons were corrected for multiple testing (e.g., using FDR). If not, some of these findings may be spurious. These should be clarified in the figure legend or methods.
Author Response
see attached word file for response to reviewer 1.

Reviewer 2 Report
Comments and Suggestions for Authors
Review of the article: "Distinct Gut Microbiota Signatures Are Associated with Severity of Metabolic Dysfunction-associated Steatotic Liver Disease in People with HIV"
The article is an important and timely study devoted to the study of the relationship between the composition of the gut microbiome and the severity of metabolic dysfunction-associated steatotic liver disease (MASLD) in people with HIV (PWH).
The work is relevant given the increasing prevalence of MASLD among PWH and the poorly understood role of the microbiome in the progression of this pathology.
The authors use modern methods, including 16S rRNA sequencing and functional analysis of the microbiota, which makes the study significant for clinical and fundamental medicine.
MASLD is one of the leading causes of chronic liver disease in PWH, and the role of the microbiome in its progression remains poorly understood. The study focuses on a high-risk group, which is important for the development of personalized approaches to diagnosis and treatment. The authors used transient elastography (FibroScan) and biomarkers (CK-18) for non-invasive assessment of fibrosis and steatohepatitis (MASH) to increase the clinical applicability of the results. Microbiota analysis was performed using 16S rRNA sequencing and PICRUSt to predict functional pathways, which meets current standards. As a result, the authors identified significant differences in the composition of the microbiota between groups with severe and non-severe MASLD. Associations of certain bacterial taxa (e.g., Ruminococcus gnavus, Prevotella) with disease severity were also found, which is consistent with previous studies, and a correlation of microbial enzymes with triglyceride levels was established. The results highlight the potential of the microbiome as a biomarker for risk stratification in PWH and open up new avenues for research, such as the impact of microbiota modification on the course of MASLD.
The limitations include the small sample size (n=34) and the lack of control groups, which would have strengthened the strengths of the article. The lack of histological confirmation. The claim that "microbial signatures can complement existing biomarkers" is questionable due to the small sample size.
Despite the limitations, the article makes a significant contribution to the understanding of the role of the microbiome in MASLD in PWH. The data obtained enrich the scientific base and open up new avenues for research, including potential therapeutic targets. The work deserves publication, but further studies are required to confirm the results.
Author Response
see attached word document for response to reviewer 2.
